# OpenReview forum: "A Game-Theoretic Analysis of Attack by Hiding Intent"
_ICLR.cc/2026/Conference — ICLR 2026 Conference Withdrawn Submission_

### Official Review · Reviewer_9ia9 · 2025-10-28

**Soundness:** 3
**Presentation:** 3
**Contribution:** 3
**Rating:** 4
**Confidence:** 4

**Summary:**

This paper proposes a novel adversarial prompting strategy in which malicious goals are concealed through skill composition. A game-theoretic framework is developed to analyze attacker-defender dynamics under prompt and response filtering, revealing structural advantages for the attacker. The authors further design a theoretically grounded defense mechanism that misleads the attacker and demonstrate its effectiveness through empirical evaluation.

**Strengths:**

1. The paper introduces a formal game-theoretic framework to model adversarial prompting via hidden intent, which goes beyond prior heuristic-based jailbreak techniques. The equilibrium analysis offers valuable theoretical insights into attacker-defender dynamics and scalability of attacks.
2. By modeling intent obfuscation through compositional skills, the paper unifies and generalizes a wide range of existing jailbreak methods under a single formalism. This abstraction helps clarify the underlying structure of complex adversarial prompts.
3. The proposed defense strategy actively misleads attackers by fabricating weak points, a concept grounded in mechanism design. Theoretically, this eliminates the attacker’s combinatorial advantage and shows provable improvement over naive capacity-based defenses.
4. The attack and defense methods are empirically evaluated on multiple LLMs with consistent performance trends. The introduction of the Bin-JR score offers a clear and reproducible way to measure real-world attack effectiveness.

**Weaknesses:**

1. The theoretical model assumes the attacker has access to accurate estimates of the defender's vulnerabilities, which may not hold in real-world black-box settings. This idealized assumption limits the practical applicability of the theoretical results.
2. While the formulation is general, the paper does not deeply analyze the structure or dependencies within the skill space. The assumption that skills are uniformly combinable and effective may oversimplify real adversarial prompting behavior.
3. The defense mechanism requires the defender to fabricate misleading responses while maintaining plausible output, which could degrade user experience or conflict with standard alignment goals.
4. Although the experiments cover multiple LLMs, some strong jailbreak baselines are missing. Additionally, the evaluation depends heavily on LLM-based raters, whose own vulnerabilities or subjectivity are not analyzed in depth.

**Questions:**

Please refer to the weaknesses.

---

> ### Author Response · Authors · 2025-11-19
>
> We thank the reviewer for the thoughtful assessment and for highlighting several key strengths of our work in terms of our theoretical framework, equilibrium analysis, and defense mechanism.
>
>
> **W1.** Our analysis characterizes the worst-case attacker advantage with accurate estimates of the defender's vulnerabilities. This is intentional and consistent with many engineering practices in the real world to ensure a safety margin, which could drive development of more robust defense.
>
>
> **W2.** Skills can have correlations. However, our theoretical analysis focuses on the worst-case behavior. For example, if the attacker presents sufficiently distinct skills, the defender will eventually spread their finite budget thinner. The focus on worst-case analysis is intentional and consistent with many engineering practices in the real world to ensure a safety margin to drive the development of more robust defense. In reality, the space of sufficiently different skills is still large and it can be combinatorially large by mixing multiple skills. The theory mainly tells us that attackers may scale their efforts relatively more efficiently than the defenders, alerting to a potential risk in worst-case scenarios.
>
>
> **W3.** Our defense mechanism is designed solely to mislead users with malicious intents, whose fulfillment is generally not regarded as yielding positive utility in common practice.
>
> **W4.** This work primarily focuses on theory. Our experiments are only complementary to our theoretical results and demonstrate a realistic threat. Achieving state-of-the-art empirical performance is not our goal.
>
>
> We acknowledge the importance of ensuring that LLM-based raters are reliable. To directly test reliability, Table 1 compares our raters’ decisions with human judgments. We observe high agreement rates (89%) and low false-positive/false-negative rates, indicating that the rater behaves consistently with human evaluators. This empirical validation shows that our conclusions do not hinge on uncontrolled subjectivity or vulnerabilities of the LLM rater. Instead, the rater provides a stable approximation of human evaluation, which is standard practice in recent safety-evaluation work.
>
>
> Besides, in Table 1, we explicitly audit multiple candidate raters on a human-labeled dataset and report their agreement, FPR, FNR, and refusal behavior. This analysis reveals that some LLMs are indeed weaker raters (e.g., high refusal rate or misalignment), and we therefore select GPT-4.1 specifically because it has the highest agreement with human experts and favorable error rates.
>
>
> The analyses show that we do not naively depend on an unanalyzed LLM rater, but instead quantify, compare, and control rater vulnerabilities before using it in our evaluation.
>
> We thank the reviewer once again for their thoughtful efforts throughout the review process. We remain eager to answer any further questions you may have.

---

### Official Review · Reviewer_HTJT · 2025-10-29

**Soundness:** 2
**Presentation:** 3
**Contribution:** 2
**Rating:** 2
**Confidence:** 4

**Summary:**

The paper introduces three things:

1. A jailbreaking strategy, Attack by Hiding Intent (AHI).
2. A specific defense mechanism for this attack.
3. A game theory model for the attacker and defender when attacker is using AHI and analysis of the equilibrium point of this game.

AHI essentially involves taking some malicious prompt with a specific intention (e.g. get information on how to make a bomb) and then mutating the prompt using a skill. The defender then sees the mutated prompt and attacked LLM output, and has to decide if they will allow it or not.

The authors formalize this attack defense game in equation (1). Critically, their analysis hinges on a defender complexity constrain $c$. The analysis requires that the defender can essentially "spend" this capacity on defense accuracy on intent skill pairs $(i, s)$. I will refer to this as the Defender Complexity Assumption (DCS).

The theoretical results in the paper hinge on DCS. In the standard defense setup, Theorem 3.1 shows that the game becomes easier for the attacker as they increase the number of skills because the defender must "spread" their finite model complexity between defending against more skills. This directly motivates their defense, in which they change the rules of the game to have a probing and attack phase. In this game, the defender can do better by misleading the attacker in the probing phase.

The paper contains some empirical results also. In Table 2 they show that AHI is an effective jailbreaking attack against the standard defender setup. In Table 3 they provide some empirical evidence that increasing the number of skills increases attack performance, as predicted by their theory. In Table 4 they show under the modified rules of the new setting their defense leads to a decrease in attacker performance, again as predicted by the theory.

**Strengths:**

## Originality

Permuting an input jailbreaking attack to make it more effective is not novel (e.g. [1]), however the specific use of modifying with a skill as done in AHI is new to my knowledge. In addition, the theoretical analysis is novel to my knowledge.

[1] Shah, Rusheb, et al. "Scalable and transferable black-box jailbreaks for language models via persona modulation." _arXiv preprint arXiv:2311.03348_ (2023).

## Quality and Clarity

The paper is well written and clear. The theoretical results and empirical results are explained well.

While I have concerns about the assumptions of the theoretical results (see next section), the idea of posing adversarial attack and defense as a 2 player minmax game that is tractable to analyze was interesting and enjoyable to read.
## Significance

I have concerns about the overall significance of the paper (see next section).

**Weaknesses:**

The paper has a number of weaknesses:

1. I think the DCS assumption is not reasonable or realistic. Firstly, while defending models do have finite capacity, this has been increasing incredibly quickly over the last five years. Frontier models now have trillions of parameters, and can be used as defense models. Secondly, the idea that even if you have a small complexity budget, this budget has to be spent in an exclusive manner across intent, skill pairs seems unreasonable. For example, if you have two very similar skills, the model does not have to spend the same capacity on defending against these as two very different skills. From this, it seems there is not an infinite number of different skills, and thus the conclusion from theorem 1, that the attacker can always win by increasing the number of skills does not seem reasonable.
2. Following on from 1, the empirical results do not back up this idea that increasing skill count can always lead to attacker success. In particular, theorem 1 would suggest that if you keep increasing the number of skills tested in Table 3, then the attack success rate will have to go to 100%. In contrast, the empirical results seem to show that while yes attack success rate increases with number of skills, the Bin-JR score is plateauing. If the assumptions of theorem 1 are correct, then the authors should be able to jailbreak any model (ideally one more robust than GPT-3.5) with 100% success rate.
3. The new defense presented is centered around there being a probing and then attack phase. This also seems unrealistic in the real world.

**Questions:**

My questions center around the weaknesses.

1. Can you explain why DCS is a reasonable assumption?
2. If possible, could you run additional results against frontier models, e.g. GPT 4.1 and 5, where you increase the number of skills and see that the ASR goes up reliably.
3. Can you explain further why the probing then attack phase assumption is reasonable.

---

> ### Author Response · Authors · 2025-11-19
>
> We thank the reviewer for the thoughtful and detailed feedback, and for highlighting the originality and clarity of our work. We are glad to know you found our work interesting and enjoyable to read. Below we will address the concerns and will revise our paper accordingly.
>
>
>
>
> **W1.** Relying on very large models for defense can increase latency, which in turn negatively affects user experience. In addition, real defenses must satisfy latency, cost, and context-length budgets. They cannot run arbitrarily many detectors, tools, or complex multi-pass pipelines per request. Skills, especially their combinations, represent a vast space of semantic transformations. Practical safety filters, however, have finite capacity and are hard to cover such a combinatorial mixture. Our empirical results demonstrate that even very large models like those in the GPT series remain vulnerable to our attack under the capacity constraints of their current safety mechanisms. In practice, many deployed LLM-based systems rely on small or moderate-sized models as safety filters due to strict constraints on latency, compute budget, power consumption, memory, and privacy requirements. This is increasingly common in real-world edge or on-device AI deployments, including cars (autonomous driving assistants), healthcare devices, medical triage systems, wearables, home assistants, robotics, and industrial IoT, where privacy or reliability demands require that the model be run locally. In these settings, the safety component is necessarily smaller. Our capacity constraint is therefore not an artificial assumption but a practically relevant abstraction of real deployments.
>
>
> Our theoretical analysis focuses on the worst-case behavior. For example, if the attacker presents sufficiently distinct skills, the defender will eventually spread their finite budget thinner.  While we don’t claim the skill space is infinite, the space of sufficiently different skills is still large and it can be combinatorially large by mixing multiple skills. The theory mainly tells us that attackers may scale their efforts relatively more efficiently than defenders, alerting to a potential risk in worst-case scenarios. The focus on worst-case analysis is intentional and consistent with many engineering practices in the real world to ensure a safety margin to drive the development of more robust defense. We will clarify and discuss this important aspect in our revision.
>
>
> **W2.** Thanks for bringing up this concern and our empirical results should be interpreted with more nuance. Theorem 1 focuses on the worst-case scenario. In our experiments, we only consider a small number of skills. The skills we use are not optimally designed to be mutually “orthogonal” or maximally difficult to defend against; they are hard to find in practice. Models like GPT-3.5 already have non-zero robustness, so we see the performance is plateauing and are far from the asymptotic regime of the theorem. We will further clarify these conditions in our revision.

---

> ### Author Response · Authors · 2025-11-19
>
> **W3.** Many existing works [1,2,3] include a probing or search/optimization stage. One of our goals is to provide a theoretical framework to capture many existing methods in the literature. In our formulation, the attacker’s strategy is represented as a conditional distribution $ p_{\mathcal{S}|\mathcal{I}} $, where $ \mathcal{I} $ denotes a set of intents and $ \mathcal{S} $ a set of skills. Many existing attacks can be reinterpreted within our framework as _fixed-skill_ attacks, wherein a single skill is applied across all intents to evade detection. Concretely, these attacks correspond to the strategy $p_{\mathcal{S}|\mathcal{I}}(s \mid i) = \mathbb{1}_{s = s^\star}$, where $s^\star$ is a skill such as affirmative instruction [4], low-resource language prompting [5], persona or role-play [6,7], or hypothetical scenarios [8].
>
>
> Our framework also accommodates _optimization-based_ attacks [1,2,3], which adaptively search for effective skills through feedback. These can be expressed as $p_{\mathcal{S}|\mathcal{I}}(s \mid i) = p(s \mid i, f)$, where $f$ denotes feedback obtained during the optimization process, reflecting the attacker's attempt to identify vulnerabilities in the defense system.
>
>
>
>
> [1] Yu, J., Shao, Y., Miao, H., & Shi, J. (2024). Promptfuzz: Harnessing fuzzing techniques for robust testing of prompt injection in llms.
>
> [2] Shah, R., Pour, S., Tagade, A., Casper, S., & Rando, J. (2023). Scalable and transferable black-box jailbreaks for language models via persona modulation.
>
> [3] Chao, P., Robey, A., Dobriban, E., Hassani, H., Pappas, G. J., & Wong, E. (2025, April). Jailbreaking black box large language models in twenty queries.
>
> [4] Wei, A., Haghtalab, N., and Steinhardt, J. (2023). Jailbroken: How does LLM safety training fail? In Advances in Neural Information Processing Systems, volume 36, pages 80079–80110.
>
> [5] Yong, Z.-X., Menghini, C., and Bach, S. H. (2023). Low-resource languages jailbreak GPT-4. [6] Zhang, Z., Zhao, P., Ye, D., and Wang, H. (2025). Enhancing jailbreak attacks on llms via persona prompts.
>
> [7] Yu, Z., Liu, X., Liang, S., Cameron, Z., Xiao, C., and Zhang, N. (2024b). Don’t listen to me: Understanding and exploring jailbreak prompts of large language models.
>
> [8] Luo, X., Wang, Y., He, Z., Tu, G., Li, J., and Xu, R. (2025). A simple and efficient jailbreak method exploiting llms’ helpfulness.
>
> **Q1.** Please refer to W1.
>
> **Q2.** As noted in W2, it is difficult to identify skills that are difficult for GPT-4.1 or GPT-5, as these large models already have well-established robustness across a wide range of skills.
>
> **Q3.** Please refer to W3.
>
> We thank the reviewer once again for their thoughtful efforts throughout the review process. We remain eager to answer any further questions you may have.

---

> ### Comment · Reviewer_HTJT · 2025-11-24
>
> **W1**:
>
> > Relying on very large models for defense can increase latency, which in turn negatively affects user experience.
>
> I agree that relying on large models can have this effect. Given the size of frontier models, adding say a 7B monitoring model would be negligible. Your argument for AI running on edge devices is more convincing. Lines 136 to 141 of the paper where you make this argument needs citations however. I am overall, however, still unconvinced by the DCS assumption.
>
> > Our theoretical analysis focuses on the worst-case behavior. For example, if the attacker presents sufficiently distinct skills, the defender will eventually spread their finite budget thinner. While we don’t claim the skill space is infinite, the space of sufficiently different skills is still large and it can be combinatorially large by mixing multiple skills.
>
> As stated above, the assumption seems not worst-case but rather unrealistic. You say on line 148 "This models a conservative scenario in which the defender cannot benefit from
> any positive transfer across skills." I thin calling this "conservative" is not accurate. Afterwards you state "This is intentional and consistent with many engineering practices in the real world to ensure a safety margin to drive the development of more robust defense." It is true that worst case analysis is used in this way, but you need some evidence that your assumptions are indeed worst case in some meaningful way, and not just entirely unrealistic. As far as I can tell, there is not evidence for this.
>
> **W2**:
>
> >  The skills we use are not optimally designed to be mutually “orthogonal” or maximally difficult to defend against; they are hard to find in practice. Models like GPT-3.5 already have non-zero robustness
>
> I think this helps make my point above. If it is impossible to find "mutually orthogonal" skills in practice, then DCS is not reasonable.
>
> **W3**:
>
> My concern here is that your improved defense requires the defender to specifically know there is a probing phase in which they can change their strategy to mislead the attacker. This does not seem realistic to me.
>
>
> ## Summary
>
> I really appreciate the time you spent to write your rebuttals. Some points have been clarified, but overall my concerns with the paper remain. Accordingly I do not think the paper is suitable for publication and will keep my score as is.
>
> With this being said, I still think there are valuable ideas in the paper that can be developed more for resubmission. In particular, the points you raised in **W3**, that your formulation is a generalization of many prior attacks is interesting. I don't think, however, that this makes the two phase defense setup realistic. Additionally, either the DCS assumption needs to be dropped or more work needs to be done to explain why this is a reasonable worst case approximation.

---

> > ### Author Response · Authors · 2025-12-04
> >
> > We thank the reviewer for the thoughtful comments and are glad to know that you found relying on very large models for defense can negatively affect latency and user experience, and the defense constraint on AI running on edge devices is convincing.
> >
> > New empirical evidence [9] shows that even the most recent frontier models remain vulnerable to a simple skill such as poetic framing, despite their large scale and significant safety alignment efforts. The persistence of jailbreaks and adversarial prompting in frontier models suggests that defensive capacity is both practically constrained and imperfect. Therefore, we think our defense's constraint is realistic and warrants careful consideration.
> >
> > [9] Bisconti, P., Prandi, M., Pierucci, F., Giarrusso, F., Bracale, M., Galisai, M., ... & Nardi, D. (2025). Adversarial Poetry as a Universal Single-Turn Jailbreak Mechanism in Large Language Models.

---

### Official Review · Reviewer_YntQ · 2025-10-30

**Soundness:** 2
**Presentation:** 1
**Contribution:** 1
**Rating:** 2
**Confidence:** 5

**Summary:**

The paper introduces an attack strategy to bypass safety alignment in LLMs using intent hiding with the goal of bypassing both input and output filters. Essentially, the attack strategy consists of an LLM generating a malicious prompt based on an 'intent' (the attack objective) and a 'skill' (some obfuscation strategy). The authors formalize the attacker and defense by introducing a min-max problem that consists of minimizing the attack success rate of the best attack. Under the assumption that the defense has limited capacity (it cannot simultaneously defend perfectly against all skills), they then show several implications including the optimal solution of the constrained min-max problem. Under the assumption that the defense can mislead the attacker, the authors then propose a defense that specifically targets their attack strategy. Finally, the authors present empirical experiments including their attack strategy and defense proposal.

**Strengths:**

- The formalization of the attacker-defense interaction as a min-max problem is interesting and provides a non-standard perspective.
- Even though it has been studied extensively already, the investigation of alignment bypass techniques remains an important and active research area.

**Weaknesses:**

- Limited novelty of attack strategy: The attack strategy -- providing a modification technique and a goal and letting an LLM generate a corresponding attack -- is very straight-forward and it appears that many similar obfuscation attacks already exist. Here are some works that appear to propose similar LLM generated attacks (based on a quick google search): https://arxiv.org/abs/2402.18104v2, https://arxiv.org/abs/2402.16717v1, https://arxiv.org/abs/2409.14729, https://arxiv.org/abs/2310.10077

- Unrealistic capacity constraint: The central theoretical assumption that defenses have limited capacity and cannot simultaneously defend against all skills is not well-justified. With a finite set of skills (in the experiments its 10), it appears entirely feasible for a defense model to handle all of them concurrently. The capacity argument might be more compelling if framed at the individual prompt level rather than at the coarse skill category level. The authors should provide empirical evidence or theoretical justification for why skill-level capacity constraints are realistic. Currently, they consider general purpose prompted defenses, but a defense (e.g., based on a encoder-only model) specifically trained against the list of skills should be capable of detecting the fixed obfuscation reliably.

- Unclear defense mechanism: The proposed defense assumes the attacker can be mislead, but its unclear how this would be implemented in practice. In the experimental evaluation, this mechanism is only simulated rather than actually deployed.

- Presentation quality issues: The paper suffers from several writing and exposition problems, including
  - inconsistent notation (e.g., using $J$ both for payoff function and judge function)
  - many theoretical parts are phrased overly complex, but lack rigour whenever it would add something (e.g., (3) assumes J is an expectation while it seem it is not in (1))
  - unnecessarily complex formulations of relatively straightforward theoretical results that could be significantly streamlined
  - unhelpful 'Discussion' paragraphs that add no value (e.g., 'The result may seem non-intuitive, yet our theoretical analysis clarifies it.')

  Improving clarity would make the contributions more accessible and make the underlying capacity assumption more salient.

**Questions:**

1. What do you see as the main distinguishing factor of your proposed attack strategy compared to existing attacks?
2. Why do you believe the capacity assumption is reasonable? And wouldn't it make more sense to phrase your capacity argument on the prompt level?

---

> ### Author Response · Authors · 2025-11-19
>
> We sincerely thank the reviewer for the detailed feedback. We are glad that you find the min-max formalization interesting and that you recognize the importance of studying alignment-bypass strategies. Below, we address each concern and clarify misunderstandings. We are working on incorporating the improvements into our revision.
>
>
> **W1.** The goal of our attack design is to provide a theoretical construction that can unify a broad range of existing methods instead of being distinct from existing methods. For example, many intent-hiding techniques discussed in prior work cited by the reviewers can be interpreted as specific instances of skills in our framework. Some of these methods also involve search processes mirroring probing for optimal skill(s) in our attack formulation. By analyzing our attack theoretically, we aim to offer insights into existing attacks. Our primary contribution is thus a general theoretical framework, rather than the introduction of a completely new attack method. We will incorporate the references cited by the reviewer in our revision to better elaborate on this point.
>
>
> **W2.** Skills, especially their combinations, represent a vast space of semantic transformations. Practical safety filters, however, have finite capacity and are hard to cover such combinatorial mixture. Our empirical results demonstrate that even very large models like those in the GPT series remain vulnerable to our attack under the capacity constraints of their current safety mechanisms. Moreover, practical defenses are typically designed to target specific skill(s) that cover a broad class of prompt instances, rather than guarding against individual prompts. This motivates us to conduct analysis at the skill level.
>
>
> Relying on very large models for defense can increase latency, which in turn negatively affects user experience. In addition, real defenses must satisfy latency, cost, and context-length budgets. They cannot run arbitrarily many detectors, tools, or complex multi-pass pipelines per request. In practice, many deployed LLM-based systems rely on small or moderate-sized models as safety filters due to strict constraints on latency, compute budget, power consumption, memory, and privacy requirements. This is increasingly common in real-world edge or on-device AI deployments, including cars (autonomous driving assistants), healthcare devices, medical triage systems, wearables, home assistants, robotics, and industrial IoT, where privacy or reliability demands require that the model be run locally. In these settings, the safety component is necessarily smaller. Our capacity constraint is therefore not an artificial assumption but a practically relevant abstraction of real deployments.
>
>
> Our experiments using only 10 skills are intended to complement our theoretical analysis by demonstrating the practical effectiveness of the attack or a realistic threat. Even with this limited set of 10 skills mixed with a test set of malicious intents, the attack successfully bypasses the safety mechanisms of advanced models like GPT-3.5 and GPT-4.1, underscoring that the threat remains unresolved. In real-world scenarios, the number of potential skills and especially their combinations with intents can be far greater for a defender to handle. We selected only a subset for experimentation, as it is infeasible to exhaustively test all possibilities. However, our theoretical analysis provides a lens to reason about a large skill space. It suggests that simply scaling defense capacity is inefficient, as it may incur substantial monetary and energy costs, with negative implications for both economic and environmental sustainability. This motivates us to design our more efficient defense mechanism.
>
>
>
>
>
>
> **W3.** This work primarily focuses on the theory. More specifically, we theoretically show the need for efficient defense mechanisms, the effectiveness and provable optimality of our proposed defense mechanism. Our goal is to provide a theoretical foundation that can inspire future research. A practical implementation of the defense mechanism is beyond the scope and budget of this work, and we leave it as a future work.
>
>
>
>
> **W4.** Thanks for pointing out the presentation issues. We are fixing them and improving the writing accordingly.
>
>
> **Q1.** Please refer to W1.
>
>
> **Q2.** Please refer to W2.
>
> We thank the reviewer once again for their thoughtful efforts throughout the review process. We remain eager to answer any further questions you may have.

---

### Official Review · Reviewer_Zp6G · 2025-11-01

**Soundness:** 2
**Presentation:** 2
**Contribution:** 2
**Rating:** 2
**Confidence:** 3

**Summary:**

This paper formalizes the dynamics between an adversary and a defense system for an LLM in a game-theoretic manner. For a specific intent, an adversary selects one or more attacking skills to craft an adversarial prompt. The defender classifies an input as benign or adversarial upon input of the adversarial prompt and model response. The authors explore two settings: the first basic setting, where a defender detects adversarial inputs to the best of its learned abilities, and a second, more complex setting, where the defender intentionally tries to mislead the attacker by selectively accepting adversarial outputs. In these two settings, the paper proves equilibrium values and conditions. Finally, the authors evaluate their theoretical results on both open and closed-source models on two jailbreaking datasets, showing the superior performance of both their attack and defense methods.

**Strengths:**

1. The authors try to formalize an area that has mostly been focused on empirical work previously, which could guide future attack and defense designs. They also present a game-theoretic setting which extends the much simpler paradigms that currently dominate safety work (e.g., iterative attacker only, or learned input filter).

2. The paper also introduces a defense system capable of misleading an attacker. While I think the specific use case for this needs to be better fleshed out, I think this idea is quite novel and warrants further consideration.

**Weaknesses:**

1. The paper does not clearly lay out assumptions, which seem to be a bit questionable. For instance, it seems it is assumed that all skills are equally "valuable" to the defender, such that you have clear trends in Theorem 1 w.r.t. $\mathcal{S}$. However, this assumption does not seem to hold well in reality, where, for instance, filtering with stronger models is more powerful than filtering with weaker models. I think it would generally be helpful if the authors included an assumptions subsection (maybe in section 2), where each assumption is clearly listed and justified as is commonplace in framework definitions.

2. In lines 152 - 155, you explain that the utility of the attack output is not considered and that the utility is assumed to be 1. I think this is quite a large problem for the analysis, as the intent is essentially meaningless, as the attacker's sole objective in this setting is to fool the defense classifier. I am also confused about why the authors then actually measure this utility empirically, as detailed in lines 302 - 308.

3. Theorem 3.2 is that the equilibrium value is maximized when the prior on $\mathcal{I}$ is uniform. I'm not sure this is a really helpful insight or one worth highlighting, since almost always in LLM systems, attacks are targeted, so the attacker will not actually sample from $\mathcal{I}$.

4. The authors create a new JRR metric. While this metric makes sense in the context of their paper, results should also be presented in more popular metrics like attack success rate (ASR).

**Questions:**

1. In equation 1, there is a constraint on $C$, however, $C$ is not explicitly mentioned in the equation. Could you clarify where it appears?

2. Most of the theory in the paper does not seem to require M to necessarily be an LLM. Where does M being an LLM mathematically fit in? Or is this a more general formalization for adversarial attacks and defenses? If so, I think the framing of the paper would need to be changed to reflect this.

---

> ### Author Response · Authors · 2025-11-19
>
> We thank the reviewer for the detailed feedback and constructive comments. We are glad that you find the formalization direction valuable and consider our misleading-defender mechanism novel. We will address and clarify your concerns below and are working on incorporating the suggested improvements into our revision.
>
>
> **W1.** Thank you for pointing this out and this is a very helpful suggestion. Assumptions are stated in the paper but currently scattered; we agree that consolidating them improves clarity.
> Our theoretical analysis focuses on the worst-case behavior. For example, if the attacker presents sufficiently distinct skills, the defender will eventually spread their finite budget thinner. The focus on worst-case analysis is intentional and consistent with many engineering practices in the real world to ensure a safety margin to drive the development of more robust defense. In reality, the space of sufficiently different skills is still large and it can be combinatorially large by mixing multiple skills. The theory mainly tells us that attackers may scale their efforts relatively more efficiently than the defenders, alerting to a potential risk in worst-case scenarios.
>
>
> In practice, many deployed LLM-based systems rely on small or moderate-sized models as safety filters due to strict constraints on latency, compute budget, power consumption, memory, and privacy requirements. This is increasingly common in real-world edge or on-device AI deployments, including cars (autonomous driving assistants), healthcare devices, medical triage systems, wearables, home assistants, robotics, and industrial IoT, where privacy or reliability demands require that the model be run locally. In these settings, the safety component is necessarily smaller. Our capacity constraint is therefore not an artificial assumption but a practically relevant abstraction of real deployments.
>
>
>
>
> **W2.** The intent is not meaningless. For example, the performance of the defender depends on the intent. From a defender’s perspective, our analysis with this assumption focuses on the worst-case of its potential risks to guide practice for ensuring safety margin. The actual utility could be subjective and hard to formulate theoretically. However, we introduced an empirical method to measure it in our experiments and used it as a secondary metric to better simulate real-world conditions beyond our theoretical worst-case analysis and to provide a comprehensive evaluation of the attacks.
>
>
> **W3.** The theorem is about worst-case vulnerability.  A uniform prior corresponds to the maximum uncertainty the defender faces. In real systems, defenders often protect against a wide range of harmful intents from an unknown distribution. For example, if an attacker wants to discredit a LLM, this theorem can help identify strategies that most effectively expose its vulnerabilities. Similarly, red-teaming methods often explore a variety of malicious intents instead of a targeted one to identify vulnerabilities of a target system.
>
>
> **W4.** Given our specific setting distinct from prior work, we define a successful attack as one that bypasses the filter and yields a response that is helpful to the attacker’s intent. In this context, our Bin-JR score can be interpreted as an ASR score.
>
>
> **Q1.** The capacity $C$ is defined in line 133, just before Equation (1). We will revise the manuscript to make it clearer.

---

> ### Author Response · Authors · 2025-11-19
>
> **Q2.** The development of our theoretical framework is motivated by common attacker–defender interactions in the context of LLMs, such as intent hiding and safety filtering. Our framework also captures many existing LLM attack methods in the literature. In our formulation, the attacker’s strategy is represented as a conditional distribution $ p_{\mathcal{S}|\mathcal{I}} $, where $ \mathcal{I} $ denotes a set of intents and $ \mathcal{S} $ a set of skills. Many existing attacks can be reinterpreted within our framework as _fixed-skill_ attacks, wherein a single skill is applied across all intents to evade detection. Concretely, these attacks correspond to the strategy $p_{\mathcal{S}|\mathcal{I}}(s \mid i) = \mathbb{1}_{s = s^\star}$, where $s^\star$ is a skill such as affirmative instruction [1], low-resource language prompting [2], persona or role-play [3,4], or hypothetical scenarios [5].
>
>
> Our framework also accommodates _optimization-based_ attacks [6,7], which adaptively search for effective skills through feedback. These can be expressed as $p_{\mathcal{S}|\mathcal{I}}(s \mid i) = p(s \mid i, f)$, where $f$ denotes feedback obtained during the optimization process, reflecting the attacker's attempt to identify vulnerabilities in the defense system.
>
>
>
>
> The model M is directly relevant to an LLM, as in our setting, skills act through semantic transformations of natural language, the input to M, rather than through low-level feature manipulations. Our defense mechanism, which misleads the attacker, is also inspired by hallucination, a characteristic behavior of LLMs. While this work primarily aims to establish theoretical foundations, we plan to incorporate more LLM-specific features into our analysis in future work.
>
> We thank the reviewer once again for their thoughtful efforts throughout the review process. We remain eager to answer any further questions you may have.
>
> [1] Wei, A., Haghtalab, N., and Steinhardt, J. (2023). Jailbroken: How does LLM safety training fail?
>
>
> [2] Yong, Z.-X., Menghini, C., and Bach, S. H. (2023). Low-resource languages jailbreak GPT-4.
>
>
> [3] Zhang, Z., Zhao, P., Ye, D., and Wang, H. (2025). Enhancing jailbreak attacks on llms via persona prompts.
>
>
> [4] Yu, Z., Liu, X., Liang, S., Cameron, Z., Xiao, C., and Zhang, N. (2024b). Don’t listen to me: Understanding and exploring jailbreak prompts of large language models.
>
>
> [5] Luo, X., Wang, Y., He, Z., Tu, G., Li, J., and Xu, R. (2025). A simple and efficient jailbreak method exploiting llms’ helpfulness. arXiv preprint arXiv:2509.14297.
>
>
> [6] Chao, P., Robey, A., Dobriban, E., Hassani, H., Pappas, G. J., and Wong, E. (2023). Jailbreaking black box large language models in twenty queries.
>
>
> [7] Zou, A., Wang, Z., Carlini, N., Nasr, M., Kolter, J. Z., and Fredrikson, M. (2023). Universal and transferable adversarial attacks on aligned language models.

---

> > ### Comment · Reviewer_Zp6G · 2025-11-27
> >
> > Thank you for your response. A few comments
> > > we agree that consolidating them improves clarity
> >
> > Can you point out the line numbers in the revised version where these are listed clearly?
> >
> > > The intent is not meaningless.
> >
> > Thank you, I think this clarification is helpful and should be added to the paper.
> >
> > > common attacker–defender interactions in the context of LLMs, such as intent hiding and safety filtering
> >
> > Would the main difference preventing this framework from being useful for other types of attacks be the intent? My original question was touching on how the paper treats LLMs as black boxes without using any LLM-specific elements of the models, like decreasing the PPL of a desired output, etc.

---

> > > ### Author Response · Authors · 2025-11-27
> > >
> > > We appreciate your feedback and are encouraged by your acknowledgement of our response.
> > >
> > > > Can you point out the line numbers in the revised version where these are listed clearly?
> > >
> > > Please refer to lines 600-623 in the Appendix. We want to thank the reviewer for this valuable suggestion, which improves the clarity of our work. Due to page limitations and to preserve space for incorporating additional reviewer feedback, we decided to place it in the appendix, but mention it in section 2 as suggested by the reviewer.
> > >
> > > > Thank you, I think this clarification is helpful and should be added to the paper.
> > >
> > > We’re glad that the clarification was helpful and have incorporated this clarification into the revised version in the discussion of the utility setup and evaluation metrics.
> > >
> > > > Would the main difference preventing … like decreasing the PPL of a desired output, etc.
> > >
> > > We are glad to further clarify this. Our framework is grounded in common LLM jailbreaking practices, where attacks operate entirely at the prompt level and treat the model as a black box. LLMs can be jailbroken by crafting prompts that bypass their safety mechanisms and elicit helpful responses toward a malicious intent, rather than random outputs, even when the original prompts are rejected. This has been demonstrated in prior work [1,2,3].
> > >
> > > To capture this LLM-specific behavior, our framework uses skills as semantic transformations of natural language to craft prompts fed into a target model, $M$, which must have a natural-language interface. The model $M$ should produce natural-language responses so that they can be semantically assessed by downstream components $D$ for filtering and $R$ for helpfulness to an intent expressed in natural language. Our simplified utility is also partly motivated by the empirical observation that frontier LLM systems are powerful enough that, once their safety mechanisms are bypassed, they are highly likely to generate helpful responses. This semantic structure is what makes the attacker’s strategy space LLM-specific. Furthermore, our proposed defense leverages hallucination, a characteristic behavior of LLMs, to mislead attackers.
> > >
> > > It remains unclear whether our framework could meaningfully transfer to a non-LLM system while still keeping every component related and capturing its real-world behaviors, especially if such a system lacks a natural-language interface, semantic skill transformations, or hallucination-like generative behaviors. Nonetheless, we agree that extending the framework to other domains is an interesting direction, but we should approach it with extra caution.
> > >
> > > We sincerely appreciate the reviewer’s thoughtful feedback and comments, which are crucial for improving our work.

---

### Note · Authors · 2026-01-19

**Comment:**

We sincerely thank the AC and reviewers for their valuable feedback. After carefully considering the reviews and in light of the very limited opportunity for discussion caused by the OpenReview technical incident, we have decided to withdraw the current submission. We plan to submit an updated work to an alternative venue.

**Withdrawal Confirmation:**

I have read and agree with the venue's withdrawal policy on behalf of myself and my co-authors.